# Semantic Segmentation of Maxillary Teeth and Palatal Rugae in Two-Dimensional Images

**DOI:** 10.3390/diagnostics12092176

**Published:** 2022-09-08

**Authors:** Abdul Rehman El Bsat, Elie Shammas, Daniel Asmar, George E. Sakr, Kinan G. Zeno, Anthony T. Macari, Joseph G. Ghafari

**Affiliations:** 1Department of Mechanical Engineering, Maroun Semaan Faculty of Engineering and Architecture, American University of Beirut, Beirut P.O. Box 11-0236, Lebanon; 2Electrical and Computer Engineering Department, Faculty of Engineering, Université Saint Joseph, Beirut P.O. Box 17-5208, Lebanon; 3Division of Orthodontics and Dentofacial Orthopedics, Faculty of Medicine, American University of Beirut, Beirut P.O. Box 11-0236, Lebanon

**Keywords:** superimposition, machine learning, artificial intelligence

## Abstract

The superimposition of sequential radiographs of the head is commonly used to determine the amount and direction of orthodontic tooth movement. A harmless method includes the timely unlimited superimposition on the relatively stable palatal rugae, but the method is performed manually and, if automated, relies on the best fit of surfaces, not only rugal structures. In the first step, motion estimation requires segmenting and detecting the location of teeth and rugae at any time during the orthodontic intervention. **Aim**: to develop a process of tooth segmentation that eliminates all manual steps to achieve an autonomous system of assessment of the dentition. **Methods**: A dataset of 797 occlusal views from photographs of teeth was created. The photographs were manually semantically segmented and labeled. Machine learning methods were applied to identify a robust deep network architecture able to semantically segment teeth in unseen photographs. Using well-defined metrics such as accuracy, precision, and the average mean intersection over union (mIoU), four network architectures were tested: *MobileUnet, AdapNet, DenseNet, and SegNet*. The robustness of the trained network was additionally tested on a set of 47 image pairs of patients before and after orthodontic treatment. **Results**: SegNet was the most accurate network, producing 95.19% accuracy and an average mIoU value of 86.66% for the main sample and 86.2% for pre- and post-treatment images. **Conclusions**: Four architectural tests were developed for automated individual teeth segmentation and detection in two-dimensional photos that required no post-processing. Accuracy and robustness were best achieved with SegNet. Further research should focus on clinical applications and 3D system development.

## 1. Introduction

A crucial step in orthodontic treatment is the assessment of tooth movement, which has been studied using imaging modalities that include photographic 2D images, 2D X-rays, and intraoral 3D scanners. A precursor to measuring tooth movement is the detection, segmentation or separation of data that represents individual teeth [1]. In this paper, we focus on the detection and segmentation of individual teeth in 2D photographic images. Individual teeth segmentation has been achieved through a geometric approach and a machine learning approach.

With the geometric approach, teeth detection was addressed through the segmentation of maxillary individual teeth in 3D intraoral scans using minimum curvature to initiate the segmentation process [2]. However, user interaction was required at multiple stages to exclude the undesirable areas picked by the curvature-based algorithm. Li et al. [3] performed curvature calculations to identify teeth from a facial smile and intra-oral scan. The process involved contour and curvature calculations followed by thresholding and boundary refinement to achieve segmentation of the front teeth, an insufficient approach to use for further calculations. Another approach involved tooth region separation in 2D X-rays by creating a dental arch represented as a four-degree polynomial [4]. This method validated separating the teeth on 2D X-rays by placing planes between the adjacent teeth; however, the teeth were not segmented. 

Trials to segment teeth using machine learning are challenging because of the high variability of teeth shapes among humans and the necessity to develop a trained network and a machine learning model capable of segmenting teeth with high accuracy. Different schemes have been used, mainly on X-rays rather than oral scans. Oktay [5] applied object detection machine learning algorithms on panoramic 2D X-ray images, whereby each tooth was labeled by a bounding box belonging to a specific class of teeth: molars, premolars, and anterior teeth (canines and incisors). Preprocessing was performed to determine regions in images where teeth were expected to be found and to define symmetry [5]. An identified tooth in the X-ray image was highlighted with an encompassing rectangular shape, but this rendition is not the actual shape of the tooth.

Mikia et al. [6] applied deep learning to classify teeth, but the input to the network consisted of the X-ray images of manually pre-segmented teeth on a limited number of 52 images, considered insufficient for training purposes. The process only classified the tooth without generating a precise segmented boundary that can be used for further calculations.

Using an Artificial Neural Network (ANN), which is typically employed in tasks involving pattern recognition in the analysis of digital images, Raith et al. [7] classified dental features from a 3D scan. The features of interest, the cusps of the teeth, were input as feature vectors. Three cusp detection approaches were compared that only classified the cusps but did not segment the individual teeth. Lee et al. [8] performed instance segmentation of teeth, gingiva, and facial landmarks limited to the frontal smile-teeth to have an accuracy above 80%. Lower accuracy was achieved for the teeth farther from the center. The method was biased towards the front teeth.

Of greater relevance was the 3D model segmentation by Xu et al. [9]. The study classified mesh faces on a two-level segmentation, the first separating the teeth from the gingiva and the second segmenting individual teeth. A label optimization algorithm was introduced after each prediction to correct wrongly predicted labels. Nevertheless, “sticky teeth” (pairs of adjacent teeth similarly labeled after optimization) were sometimes falsely predicted. This problem was corrected with Principal Component Analysis. Finally, the predicted labels of the second segmentation were back-projected to the original model. This method required three-dimensional scan data and involved pre- and post-processing steps of the input to generate a precise prediction. Our method aims to generate automated predictions without any additional steps being applied to the input while generating the prediction.

One of the most significant goals of reproducing the dentition during growth or orthodontic tooth movement is the ability to determine tooth displacement relative to fairly stable oral structures, such as the palatal rugae, obviating the need to take sequential harmful radiographs. Present methods of rugae superimposition are performed manually and, if automated, rely on the best fit of surfaces, not only rugal structures. In a first step, motion estimation requires segmenting and detecting the location of teeth and rugae at any time during the orthodontic intervention.

Considering that the available methods of tooth prediction relied on manual manipulation or did not involve tooth segmentation, we aimed to generate automated segmentation consistent with all the teeth as well as precise predictions without any manual steps applied to the input while generating the prediction. In this paper, we present the development of tooth segmentation using readily available 2D occlusal photographs of the maxillary dental arch through a process that eliminates all manual steps to attain a completely autonomous system. Different architectures are tested in this process to find the most suitable network for the investigated dataset. A secondary aim was to compare four architectural styles that we planned on using.

## 2. Material and Methods

The study included the creation of a labeled dataset of photographic images of actual patients taken at the occlusal view of the maxillary arch. Deep learning was implemented through a Fully Convolutional Neural Network (F-CNN) architecture to semantically segment individual teeth and palatal rugae in color images. A benchmark using different network architectures was performed to assess the effect of data augmentation on the semantic segmentation of teeth. Accuracy and associated metrics were defined to identify the best network architecture for semantic segmentation of maxillary teeth and palatal rugae.

### 2.1. Dataset Collection

The dataset consisted of 797 photographic images compiled by the Division of Orthodontics and Dentofacial Orthopedics at the American University of Beirut Medical Center (AUBMC). In contrast to previous publications in which X-ray images were used, colored RGB two-dimensional (2D) images of the maxillary teeth and palate belonging to various malocclusions were taken according to standards at the occlusal plane with a single-lens reflex camera using an intraoral mirror. The images were taken at different distances to generate diversity in the dataset and make the training more robust. Images from the late mixed to permanent dentitions were included to illustrate the differentiation between primary canines and molars from permanent canines and molars. Also included to widen the scope of training were images with dental appliances, significant crowding, and prosthesis (including primary teeth with stainless steel crown) (Figure 1). Excluded were images with multiple missing teeth, supernumerary, and transposed teeth. The images were saved in “PNG” format and with a 480 × 320 pixel resolution because this was the common resolution for the majority of the taken images and required less memory for training compared to high resolution images.

A total of 719 images were segmented into 5 families of anatomical structures, including 4 for teeth (molars, premolars, incisors, canines) and one for the rugae. This set of data contributed to training a network to semantically segment these families (Figure 2A–C). To test individual structures, we augmented the sample with 78 randomly selected images. A total of 797 images were segmented into individual structures, comprising 23 labels for individual primary (*n* = 10) and permanent (*n* = 12) teeth as well as the rugae (*n* = 1) (Figure 2D,E).

An additional dataset composed of 47 pairs of images was utilized *solely* to test the robustness of the trained network. Each pair of images was taken of the same patient before and after orthodontic treatment (Figure 3).

### 2.2. Dataset Labeling Methods

Image labels serve as the ground truth for the training, validation, and testing of various neural network architectures. Labeling for semantic segmentation consists of assigning a class to every pixel in an image. In this work, the users who performed the labeling (orthodontic residents) identified the pixels in an image in the form of polygons drawn to fit the shape of the object of interest. The labeling was applied to the entire dataset of (797 + 2 × 47 = 891) images.

#### 2.2.1. Semantic Labeling

The pixel labeling was performed in a MATLAB application [10] by manually creating polygons following the contour of the regions of interest. For each image in the dataset (Figure 2A), an associated image of similar dimensions was created to identify the labels by assigning a different color to each label (Figure 2B,E). The teeth were captured upon superimposing the label on the original image (Figure 2C,F). All labels and contours of the teeth and the rugae area were verified by the labeling orthodontists. 

#### 2.2.2. Label Statistics

Semantic segmentation of teeth and rugae is challenging because of the variable sizes of the labels. The relatively larger size of the rugae label compared with the labels of individual teeth could skew the results of the network training. This issue was mitigated by utilizing an appropriate training accuracy metric (see Section 2.3.3. below). The number of pixels associated with each label was tallied for each labeling scheme and dataset combination.

### 2.3. Machine Learning and Semantic Segmentation

Deep learning is a model designed to analyze data similar to human analysis by using a layered structure of algorithms called an artificial neural network [11]. The design of such networks was inspired by the biological neural networks of the human brain. The algorithms are trained to find and identify patterns and features in massive amounts of data, enabling the network to generate predictions.

To apply semantic segmentation machine learning, we identified a labeled dataset comprised of images and their associated labels. The labeled images were fed into a chosen network with a specific architecture as input (see Section 2.3.2). The network in turn produced a prediction of the label as an output. To assess the accuracy of the network, the predicted label was compared to the input label, which is considered as ground truth. While in most instances, the training process starts with initial values referred to as a pre-trained network, such as the semantic segmentation application used by Siam et al. [12,13], in our study, the main architecture was trained from scratch because of the absence of a pre-trained network relevant to teeth segmentation.

#### 2.3.1. Dataset Split

In typical machine learning applications, the data are split into three categories: training (in which most of the data are used), validation (against which the training progress is validated), and testing. The validation and test sets are disjoint from the training set. Nearly 90% of our dataset with the family of teeth labeling scheme were dedicated as training and validation sets (Table 1), which were further split into 92% for training (586 images), and 8% for validation (53 images). Likewise, nearly 90% of the dataset with the individual teeth labeling scheme were earmarked for training and validation, which were also further split into 91% for training (641 images) and 9% for validation (64 images). The data from the 47 pairs of before and after images of patients were entirely used for testing to assess the robustness of the trained model.

#### 2.3.2. Network Architectures

We applied four architectures (described in Appendix A) because they would cover various characteristics of our dataset that none of them would singularly:The MobileUnet network [14], comprised of a small number of layers, is hence relatively fast to train and widely used in medical applications;The AdapNet network [15] designed to adapt to environmental changes and focus less on the environment when predictions are made. Images taken in different lighting conditions would not affect the predictive ability of this network. This feature was appropriate for our dataset because the images were taken at varying proximity;The DenseNet network [16], a model that uses features of various complexity levels to predict smooth boundaries, enables the network to deal with datasets comprising a relatively small number of images, as in our study, compared with datasets of up to hundreds of thousands of images;The SegNet network [17], designed to be efficient while using a limited amount of memory and primarily designed to perceive spatial-relationships such as road scenes, was important for our dataset that included variable distances from which images of the teeth were captured.

#### 2.3.3. Network Assessment

Semantic segmentation predictions are typically evaluated using an average mean intersection over union (average mIoU). For semantic segmentation, given two image labels representing ground truth and its associated prediction, the IoU for a given class (*c*) could be defined such that:$$IoU\left(c\right)=\frac{\sum_{i}o_{i}==c\;\cap y_{i}==c}{\sum_{i}o_{i}==c\;\cup y_{i}==c}$$
where *o_i_* is the predictions pixels, *y_i_* is the ground truth labels pixels, ∩ is a logical “and” operation, and ∪ is a logical “or” operation. This computation, visually represented in Figure 4A, is similar to the originally defined equation [18]. The mean IoU (mIoU) is an average of all the IoU values for all the classes in an image pair. For a dataset comprised of several image pairs, the average mIoU is the average of the mean IoU’s for all image pairs. 

In addition, other metrics were computed to assess the accuracy of the trained model, namely the pixel accuracy and pixel precision. Pixel accuracy is the percentage of pixels in an image that are correctly classified with respect to the input ground truth pixels.

For semantic segmentation, given two images representing the ground truth and its associated prediction, the pixel accuracy for a given class (*c*) could be defined such that:$$Accuracy\left(c\right)=\frac{\sum_{i}o_{i}==c\;\cap y_{i}==c}{\sum_{i}\;\;\;y_{i}==c}$$
where *o_i_* is the predicted pixel, *y_i_* is the ground truth label pixel, ∩ is a logical “and” operation, and ∪ is a logical “or” operation (Figure 4B).

This accuracy measure can be evaluated for a specific class in an image, as an average for all classes in an image, or as an average for a single class for the entire dataset. The latter metric is referred to as the “per-class” pixel accuracy, which can provide more information on the ability of the network to precisely segment a specific label. This metric is especially useful for classes that occupy small regions in the image.

Pixel precision is defined by the ratio of the correctly detected pixels to all predicted pixels. This metric describes how many correct predictions there are compared to the total predictions generated by the model. Similarly, given two images representing ground truth and its associated prediction, one can define the precision for a given class (*c*) such that:$$Precision\left(c\right)=\frac{\sum_{i}o_{i}==c\;\cap y_{i}==c}{\sum_{i}o_{i}==c\;}$$
where *o_i_* is the prediction pixel, *y_i_* is the ground truth label pixel, ∩ is a logical “and” operation, and ∪ is a logical “or” operation (Figure 4C).

Because the dataset exhibited a class imbalance with the dissimilar class sizes, the average mIoU is a better metric to assess the network prediction accuracy than the pixel accuracy or pixel precision. The class imbalance is exaggerated by the fact that the background (gingiva and non-teeth regions) and the rugae labels cover relatively larger areas than the rest of the classes (teeth); hence, as expected, the high pixel accuracy does not translate into a more accurate semantic segmentation [19].

One of the methodological approaches employed in the individual teeth labeling scheme involved data augmentation. We list this method and the corresponding figure under Results (Section 3.1.3) because of the sequence-dependent results.

## 3. Results

### 3.1. Machine Learning Accuracy

#### 3.1.1. Labeling Statistics

The statistics of the family and individual structure labeling schemes demonstrate the dominance of the rugae labels. The number of pixels associated with primary teeth and third molars were insignificant in comparison with the family of teeth or individual teeth. Accordingly, the accuracy in segmenting the primary teeth and the third molars was expected to be low. In the 47 pairs of (pre- and post-treatment) images (Figure 5C), the distribution of the teeth labels was similar to the training dataset (Figure 5B).

#### 3.1.2. Family of Teeth Labeling Scheme

Of the four network architectures, the SegNet and DenseNet exhibited the highest accuracy in terms of average mIoU (55.99% and 54.95%, respectively) (Table 2). For both networks, the predicted labels exhibited spatial shifts (Figure 6), indicating that the network model memorized the spatial position of the teeth rather than segmenting them. To mitigate this issue, data augmentation was employed.

#### 3.1.3. Individual Teeth Labeling Scheme

Considering the result of the family of teeth scheme, whereby the networks exhibit spatial memory, and the need for data augmentation to improve the network’s accuracy, two data augmentation methods were employed. The first involved rotating the images (and their labels) and then adding them to the original set (Figure 7). The second targets changing the perspective of the images (and their labels) by shearing them (Figure 7E).

To assess the effect of the data augmentation, the top two performing architectures, SegNet and DenseNet, were re-trained on the full dataset (Table 1). The two sets were tested individually and incrementally, culminating in six training combinations (Table 3). The highest average mIoU was on the dataset that used the rotation data augmentation only. The perspective data augmentation did not improve training accuracy. Accordingly, only the rotation data augmentation was used in the dataset for the final training. Consequently, the four original architectures were retrained on the full dataset (including the rotation data augmentation) using the individual teeth labeling scheme.

SegNet remained the best architecture, with an average mIoU of 86.66% and an accuracy of 95.19% (Table 4). A sample of the results from the SegNet architecture performed on the test dataset with the rotation augmented dataset is illustrated in Figure 8, showing the worst, average, and best predictions.

### 3.2. Machine Learning Robustness

The Signets network was used to test the robustness of the trained model on the independent third dataset of 47 pairs of pre- and post-orthodontic treatment images.

#### 3.2.1. Network Accuracy

Using the same statistical analysis on this dataset as on the previous two datasets, the number of pixels per class was computed. A sample result for the Right Central Incisor class is shown in the appendix (Figure A1A). The primary teeth and the third molar classes had the fewest pixels (close to 10% in all the images as shown in Figure A1O–T). Accordingly, two average mean IoU’s were computed, one including all classes and the second ignoring the low-occurrence classes. To focus on the teeth segmentation accuracy of the network, the rugae class was not included in the second computation of the average mean IoU, which we refer to as the “Teeth Only IoU” (Table 5).

The teeth only IoU value was 86.2 % (Table 5), but the average mean IoU of all teeth (including primary and third molars) and rugae was lower (82.9%), as expected, because the rugae boundaries were not consistently defined during the labeling process.

#### 3.2.2. Network Robustness

To validate the robustness of the trained network, the accuracy of prediction was gauged for the pre- and post-treatment images separately, which were compared in the chi-square goodness of fit test. The prediction values in both sets were not statistically different (Table 5).

## 4. Discussion

The main contribution of this study was the segmentation of 2D clinical images through artificial intelligence and machine learning to quantify orthodontic tooth movement. To the best of the authors’ knowledge, prior reporting on semantic segmentation for teeth was not available, prompting the investigation of multiple architectures to determine the best performing method. 

The goal of the segmentation was to label each pixel of the intraoral 2D image with the corresponding tooth structure. While the trained network was able to depict the spatial location of a family of teeth in the image, the initial step of segmentation was reinforced through data augmentation, resulting in a more precise estimation of a matching set of teeth. The perspective data augmentation did not improve training accuracy, possibly because the images already had perspective variability since they were taken with actual cameras.

The augmentation method is commonly used in the learning phases of segmentation to increase matching accuracy [19]. In this study, this approach induced a substantial increase in the average mIoU accuracy with the best performing architecture, SegNet, which yielded mislabeling of only 1/20th of individual tooth structure.

The rugae label exhibited the lowest accuracy, probably because the rugae area was not well-defined like the teeth. This finding was accentuated by the facts that the variation of the rugae class may have also been associated with the performance of the ground truth labels by several orthodontists, and that the defining boundary can vary between individuals. A separate investigation of the source of variation is warranted to determine the contribution of each of these factors to the rugal delineation. Operator errors in segmenting the various categories of teeth could also be explored, although in this initial project, errors in the delineation of tooth contours did not impact machine learning because tooth labeling was not affected. In addition, outliers such as those in the first row of Figure 8 show how far the worst result is from the average, demonstrating that many more such images should be included in future research for the model to be sufficiently trained on them.

The sample image exhibiting missing teeth was mislabeled by the trained model. Specifically, if one of the premolars was missing, the existing premolar could be mislabeled. For instance, in Figure 9B, the left premolar (situated on the right side of the image) was labeled correctly as the first premolar and colored in blue, while the right premolar on the opposite side was labeled incorrectly as a second premolar colored in brown. Orthodontists can label these teeth correctly because the small space between the left canine and premolar indicates the prior existence of a premolar that was extracted. The prediction of both teeth is depicted in Figure 9C. The right premolar was predicted correctly owing to the spacing; however, the left premolar was predicted falsely and was classified as a second premolar.

The lack of statistical significance between pre- and post-treatment image predictions indicates that our model was robust regardless of whether the input images were taken from the pre- or the post-treatment set. This finding reflects a valuable attribute of the trained model because most of the pre-treatment images had malaligned and crowded teeth, yet the trained model was able to correctly segment them.

Considering the high scores in accuracy and robustness, the training of the system was proven to be successful in segmenting the teeth. This achievement sets the path to validate the superimposition of images aiming to quantify tooth movement during and after the completion of orthodontic treatment. Future research should help determine stable structures or planes to superimpose images taken at different timepoints and compare them to current radiological superimposition methods for cross-validation. Thus, radiological superimposition to evaluate tooth movement would be disregarded and radiation exposure reduced.

The present two-dimensional teeth segmentation has validated the usage of machine learning tools to identify and accurately segment teeth in 2D photographs. However, the 2D imaging modality restricts the motion estimation of teeth, which is limited to planar motion (x and y) and single rotation (with respect to z), both computed in the plane of the 2D image. Hence, the estimated motion would be a projection of the actual 3D motion of the teeth. To minimize the loss of information due to projection, the next step is to apply machine learning methods to the 3D domain through estimating the 2D motion on several independent image planes or using machine learning methods on a 3D imaging modality such as intraoral 3D scans. The applicability of the 2D machine learning methods developed in this work on 3D intraoral scans should be investigated.

## 5. Conclusions

A semantically labeled maxillary teeth dataset taken at the occlusal view was used to develop autonomous tooth segmentation through a process that eliminated manual manipulation. The dataset consisted of colored images, in contrast to previous research in which X-ray images were used. Machine learning methods were applied to identify the best network architecture for semantic segmentation of the images. The best network was SegNet, which yielded 95.19% accuracy and an average mIoU of 86.66%. The developed method required no post-processing nor pre-training. The model robustness, verified on an independent set of pre- and post-orthodontic treatment images, yielded an average mIoU value of 86.2% for the individually tested teeth. This model should help develop the superimposition schemes of sequential occlusal images on stable structures (e.g., palatal rugae) to determine tooth movement, obviating the need for harmful radiation exposure.

## Figures and Tables

**Figure 1 diagnostics-12-02176-f001:**
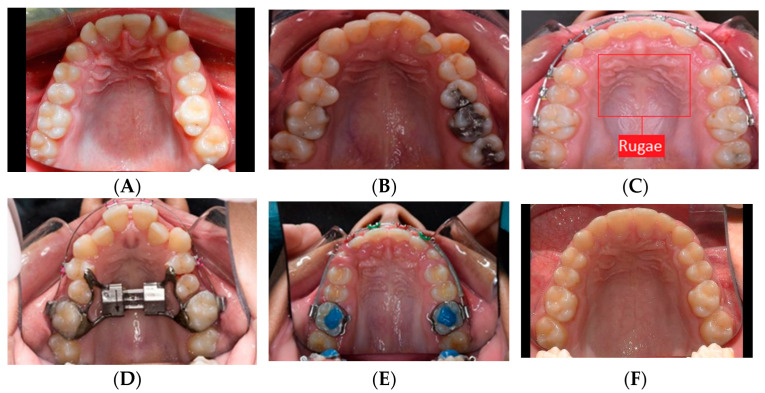
Samples of the maxillary arch images showing the variety of malocclusions, including pretreatment (**A**,**B**), during treatment (**C**–**E**), and post-treatment (**F**) images. To retain the aspect ratio of the original images, a black border was added to avoid side stretching to match the 480 × 320 size (**A**,**F**). The background change in the image did not affect the results. A sample of the selected rugae area is shown in (**C**).

**Figure 2 diagnostics-12-02176-f002:**
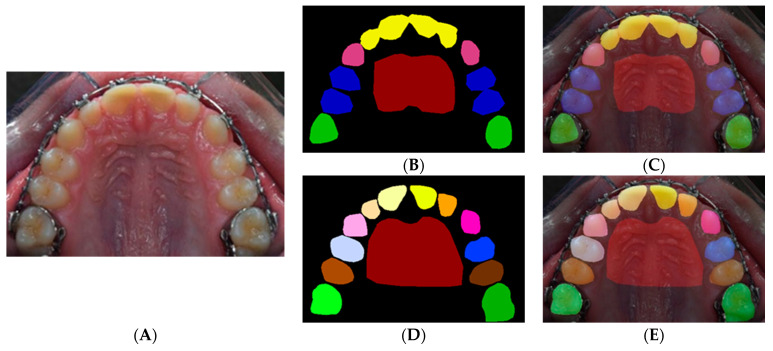
Labeling for semantic segmentation. The original image (all permanent teeth) (**A**); family segmentation: (**B**,**C**); family segmentation showing incisors (yellow), canines (pink), premolars (blue), molars (green), and rugae (maroon) (**B**); label superimposed on original image (**C**); individual teeth segmentation (**D**,**E**); a different color is assigned to every tooth (**D**); label superimposed on original image (**E**).

**Figure 3 diagnostics-12-02176-f003:**
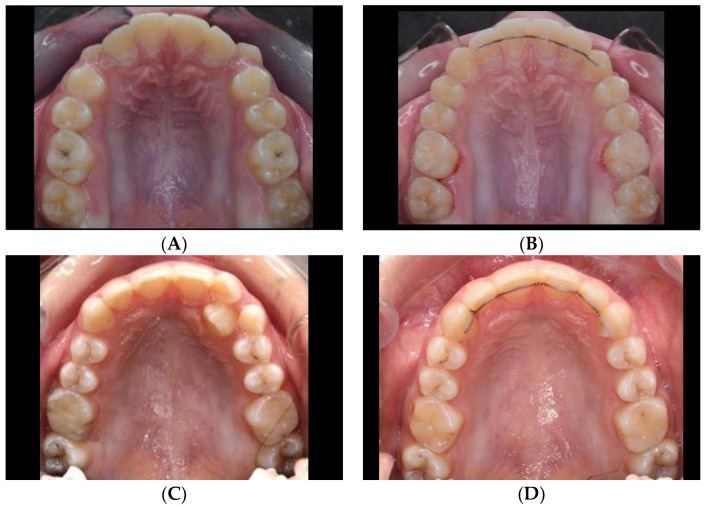
Occlusal views of the maxillary arches of treated patients with corresponding pre-treatment (left column, **A**,**C**) and post-treatment (right column, **B**,**D**) images. A total of 47 such pairs were used to test the robustness of the trained network to identify the teeth after their positions were altered through orthodontic treatment.

**Figure 4 diagnostics-12-02176-f004:**
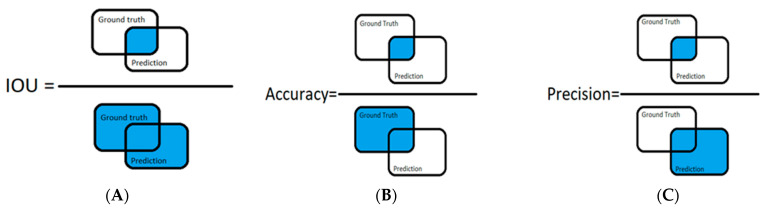
IoU (**A**), Accuracy (**B**), and Precision Representation (**C**).

**Figure 5 diagnostics-12-02176-f005:**
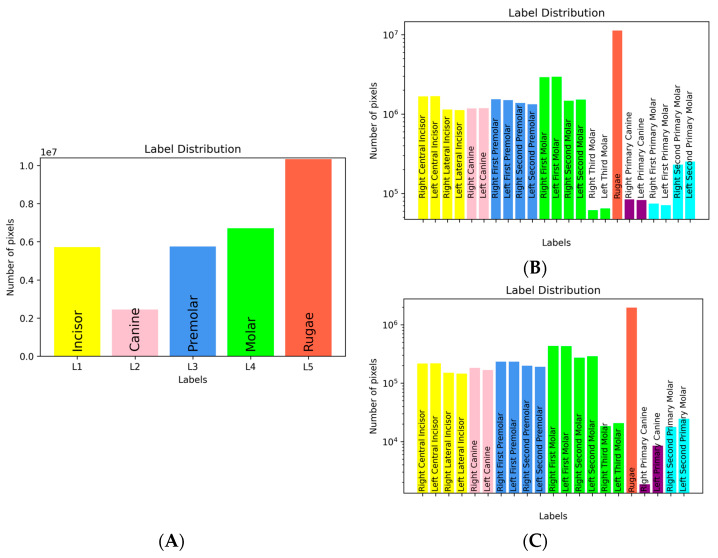
Statistics of the labeling schemes: Family of teeth labeling scheme: the rugae labels are the highest, whereas canine labels are the lowest, while those of the molars, premolars, and incisors were comparable (**A**); labeling scheme for individual adult teeth: the number of rugae pixels is greatest, and those associated with individual adult teeth are comparable (**B**); distribution of paired teeth labeling (**C**).

**Figure 6 diagnostics-12-02176-f006:**
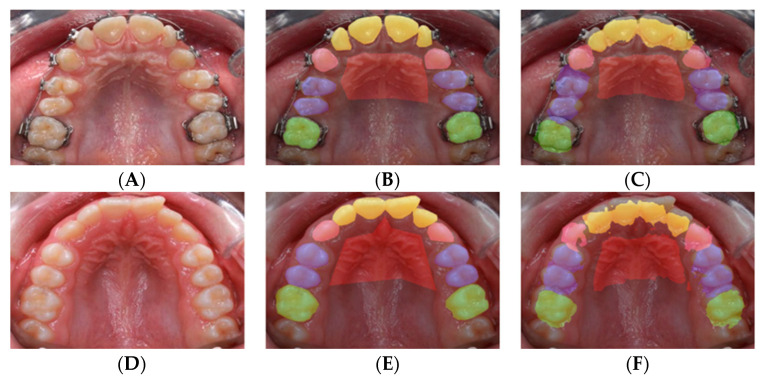
Family of teeth/rugae average sample result for DenseNet, (**A**–**C**); Family of teeth average sample result for SegNet (**D**–**F**); original images (**A**,**D**); image labels (**B**,**E**); image predictions (**C**,**F**).

**Figure 7 diagnostics-12-02176-f007:**
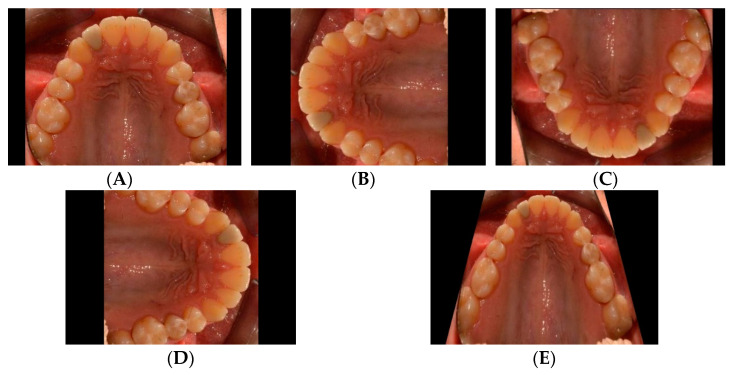
Data augmentation via rotation and perspective shrinking: Original image (**A**); 90° rotation (**B**); 180° rotation (**C**); 270° rotation (**D**); perspective (**E**).

**Figure 8 diagnostics-12-02176-f008:**
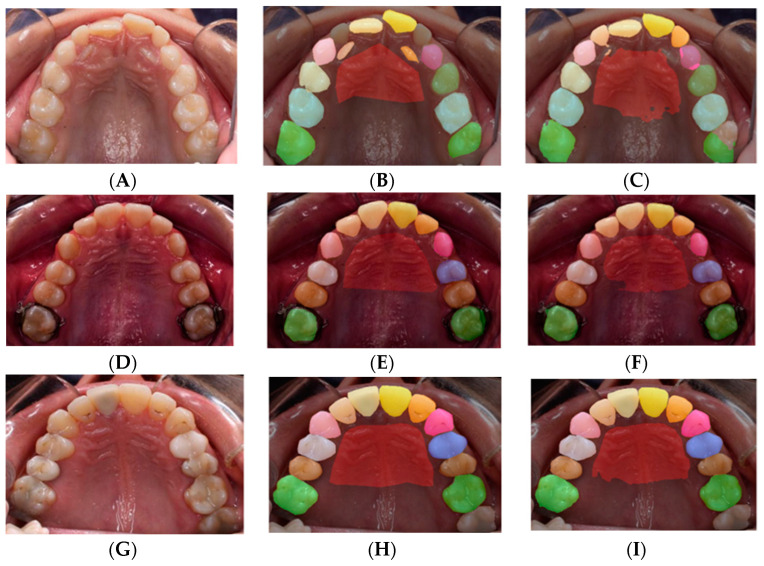
Individual Teeth Labeling results for SegNet including rotation dataset: original images (**A**,**D**,**G**); image labels (**B**,**E**,**H**); image predictions (**C**,**F**,**I**). In the first row (**A**–**C**) is the worst prediction is shown because the operator labeled the malaligned permanent lateral incisor, discarding the primary lateral (**B**), whereas the machine recognized the latter (**C**); in the second row (**D**–**F**) an average prediction is shown; the best result is shown in the third row (**G**–**I**).

**Figure 9 diagnostics-12-02176-f009:**
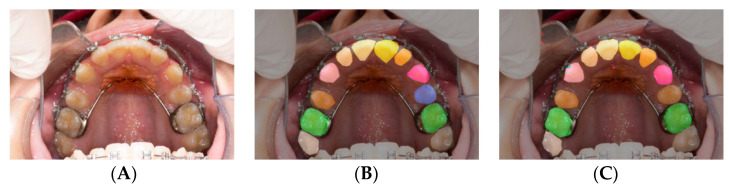
Mislabeling due to missing teeth: The original image (**A**); the ground truth label (**B**); the image prediction from the model (**C**).

**Table 1 diagnostics-12-02176-t001:** The image count and labeling scheme of the original and expanded datasets.

	Train	Validate	Test	Total
Family of teeth labeling scheme				
	Semantic Segmentation			
Count	586	53	80	719
Percent	82%	7%	11%	100%
Individual teeth labeling scheme				
	Semantic Segmentation			
Count	641	64	92	797
Percent	80%	8%	12%	100%

**Table 2 diagnostics-12-02176-t002:** Model architecture comparison on tooth family labels.

Model	MobilUNet Skip	AdapNet	FC-DenseNet56	SegNet
Class	Per-class accuracy			
Incisor	69.15	69.7	70.23	71.32
Canine	51.65	49.59	56.14	55.66
Premolar	66.76	57.76	67.36	70.36
Molar	66.89	53.17	59.54	66.02
Rugae	80.17	81.12	82.92	82.67
Void	88.15	90.88	89.25	88.47
Average accuracy	82.63	83.43	83.50	83.49
Average precision	82.80	84.58	83.80	83.47
Average mean IoU score	53.68	53.23	54.95	55.99

**Table 3 diagnostics-12-02176-t003:** Model architecture comparison on specific tooth labels.

Model	Avg. Accuracy	Avg. Precision	Mean IoU Score
DenseNet	81.84	82.41	45.89
SegNet	82.32	81.93	49.53
DenseNet-rotated	95.00	95.23	85.40
SegNet-rotated	**95.19**	**95.40**	**86.66**
DenseNet-rotation and perspective	94.68	94.77	84.19
SegNet-rotation and perspective	65.09	81.28	7.90

**Table 4 diagnostics-12-02176-t004:** Model architecture comparison on specific tooth labels with rotation for all architecture candidates.

Model	MobilUNet Skip	AdapNet	FC-DenseNet56	SegNet
Right Central Incisor	93.45	93.27	94.12	93.75
Left Central Incisor	93.04	92.48	94.14	94.39
Right Lateral Incisor	91.02	90.99	91.42	91.61
Left Lateral Incisor	91.53	91.09	93.24	92.28
Right Canine	92.70	91.78	92.06	92.36
Left Canine	91.62	90.81	92.44	92.73
Right 1st Premolar	93.19	92.88	93.28	94.18
Left 1st Premolar	92.36	91.96	94.23	93.25
Right 2nd Premolar	89.84	90.87	88.86	91.49
Left 2nd Premolar	90.99	90.06	93.04	92.03
Right 1st Molar	94.20	92.80	93.29	94.58
Left 1st Molar	94.46	90.70	92.72	92.62
Right 2nd Molar	92.66	91.32	93.03	92.39
Left 2nd Molar	92.30	91.00	93.55	94.46
Right 3rd Molar	96.70	97.38	96.76	97.22
Left 3rd Molar	95.74	95.90	95.51	96.20
Right Primary Canine	97.83	99.77	98.09	99.68
Left Primary Canine	98.26	97.99	96.69	98.84
Right 1st Primary Molar	98.73	98.64	98.70	99.23
Left 1st Primary Molar	96.77	97.26	97.08	98.54
Right 2nd Primary Molar	98.76	98.17	98.61	99.58
Left 2nd Primary Molar	98.60	97.40	97.55	98.81
Rugae	89.36	88.08	88.65	88.77
Void	96.68	96.82	97.17	97.07
Average accuracy	94.82	94.55	95.00	**95.19**
Average precision	95.03	94.76	95.23	**95.40**
Average mean IoU score	84.92	84.42	85.40	**86.66**

**Table 5 diagnostics-12-02176-t005:** Dataset analysis results.

	Average mIoU Values		
Teeth		Pre-Treatment	Post-Treatment
Right Central Incisor	86.2%	86.9%	85.5%
Left Central Incisor	85.8%	86.0%	85.6%
Right Lateral Incisor	84.7%	84.5%	84.9%
Left Lateral Incisor	82.6%	82.2%	83.0%
Right Canine	86.5%	84.5%	88.5%
Left Canine	81.6%	78.4%	84.7%
Right 1st Premolar	90.6%	89.3%	92.0%
Left 1st Premolar	90.9%	90.7%	91.2%
Right 2nd Premolar	87.0%	87.5%	86.5%
Left 2nd Premolar	86.7%	89.3%	84.2%
Right 1st Molar	91.6%	91.8%	91.4%
Left 1st Molar	89.1%	87.5%	90.6%
Right 2nd Molar	81.4%	79.1%	83.5%
Left 2nd Molar	83.2%	78.1%	87.6%
Rugae	78.6%		
Average accuracy	93.8%		
Average precision	94.3%		
All teeth and rugae average mIoU	82.9%		
Teeth only average mIoU	86.2%		

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
