# Peer review of "Semantic Segmentation of Maxillary Teeth and Palatal Rugae in Two-Dimensional Images"

_diagnostics, 2022, doi:10.3390/diagnostics12092176_

Round 1

Reviewer 1 Report

The contents of this issue are useful for dental diagnosis. I hope that this paper will be published. However, there are many points that need to be improved. In particular, the structure of each section.

Introduction:

The references are well detailed but need to be simplified. For example, Reference 4 reports a method of detection from panoramic radiographs rather than from photographs. I think it is not necessary to go into as much detail as this.

l. 68-104; these appear to be general reviews of methodology.

l. 80, 91-104; Is this an important statement to introduce the paper?

They should be rewritten with a focus on the results of previous studies, room for improvement, and reasons why a new semantic segmentation is needed.

The legend in Figure 4 should be revised to describe the key points.

Methods:

There are many sentences that should be described in the results or discussion (e.g., l. 184-189, 199-215, 218-223).

For C2 (l. 234), the reasons for choosing the four architectures used in this study should be stated in the introduction. Also, one of the aims of this study would be to compare the results of these architectures.

Results:

Figure 8 should be shown to explain the methodology.

l. 331-333, 340, 375; this is speculation or discussion, not a result. They should be listed in the appropriate section.

Discussion:

l. 440-443; These should not be described in this paper. If necessary, they should be replaced with "The next step is to apply machine learning methods to 3D..."

The first section of the abstract should also be rewritten.

Reviewer 2 Report

This paper deals with an interesting topic, but there are some concerns:

- "480x320 pixel resolution": This is rather low, compared to the digital images that are acquired clinically. However, it is understandable that high resolution images cannot be used in deep learning, due to high memory requirements. The authors should comment on this.

- Figure 1: This is not really needed. Figure 2 should be enough to demonstrate the type of images and the range.

- Figure 2: The aspect ratio of the images does not seem to be the same. Some images have an added black border on the sides, presumably to achieve the 480x320 size. Could this have affected the results?

- What was the age range of the patients? What was the range of dental age?

- The inclusion and exclusion criteria are not clearly stated. Were images with fixed or removable prostheses included? What about supernumerary teeth, missing teeth, malformed teeth (e.g. peg laterals)? What about severe crowding, transpositions, ectopic teeth?

- "A total of 797 images were acquired, of which 719 images were segmented": How were these selected?

- "All labels and contours of the teeth and the rugae area were verified by the labeling orthodontists": How many orthodontists labelled the images? What was the inter- and intra-observer error?

- Figure 7: Please report what each column shows. I assume that the central column is the ground truth, but this is not stated.

- Figure 9: It is obvious that the results of the first row are outliers. Here, both the deciduous and permanent lateral incisors are present, and the model was not trained enough on images of this kind. This is an image selection problem; such images should either not be included (and the model assumed not capable to deal with them), or many more such images should be included, so that the model could be sufficiently trained on them.

- Figure 9, first row: it is not clear whether the model correctly identified the canines and incisors as deciduous teeth. Please report any such errors, because the colour codes are not easy to see.

- Please use "premolar" instead of "bicuspid".

- The references seem too few for this topic. For example, the following relevant papers are not included:
-- Li M, Xu X, Punithakumar K, Le LH, Kaipatur N, Shi B. Automated integration of facial and intra-oral images of anterior teeth. Comput Biol Med. 2020 Jul;122:103794. doi: 10.1016/j.compbiomed.2020.103794.
-- Lee S, Kim JE. Evaluating the Precision of Automatic Segmentation of Teeth, Gingiva and Facial Landmarks for 2D Digital Smile Design Using Real-Time Instance Segmentation Network. J Clin Med. 2022 Feb 6;11(3):852. doi: 10.3390/jcm11030852.

- The reference list needs editing. For example, what is "Balancingtheregularizationeffectofdataaugmentation"?

Round 2

Reviewer 1 Report

This article is very well improved.

I hope this research will develop into the next step. Thank you.